# Acceptability and feasibility of pre-exposure prophylaxis for bacterial STIs: a systematic review

Julie-Anne Carroll[1], Amy B. Mullens[2], Sarah Warzywoda[3,4], Philip R.A. Baker[1,4,5,6‡], Meika Stafford[1‡], Faye McMillan[7‡], Jacintha Manton[7‡], Daniel Demant[7,1*]

1 School of Public Health and Social Work, Faculty of Health, Queensland University of Technology, Brisbane, Queensland, Australia, 2 School of Psychology and Wellbeing, Centre for Health Research, Institute for Resilient Regions, University of Southern Queensland, Ipswich, Queensland, Australia, 3 School of Public Health, Faculty of Medicine, The University of Queensland, Brisbane, Queensland, Australia, 4 Centre for Health Research, Institute for Resilient Regions, University of Southern Queensland, Ipswich, Queensland, Australia, 5 Australian Centre for Health Law Research, Queensland University of Technology, Brisbane, Queensland, Australia, 6 Faculty of Medicine, Universiti Teknologi MARA, Sungai Buloh, Malaysia, 7 School of Public Health, Faculty of Health, University of Technology Sydney, Sydney, New South Wales, Australia

☯ These authors have contributed equally to this work.
‡ AB and CD also contributed equally to this work.
* Daniel.demant@uts.edu.au (DD)

## Abstract

### Background

A recent resurgence of bacterial sexually transmitted infections (STIs) is placing a major burden on high-risk populations, physicians, and the healthcare system. Treatment in the form of antibiotic pre-exposure prophylaxis (STI PrEP) is a potential solution. However, little is known about the acceptability and feasibility of this approach in high-risk populations.

### Methods

A comprehensive search strategy was developed and executed in October 2024 across six databases adhering to PRISMA guidelines.

### Results

Eight studies met the inclusion criteria. These studies were all conducted in high-income countries, used various methods, and all focussed on sexual minority men. Findings consistently identified moderate to high levels of acceptability among GBMSM (54.3% - 67.5%). Factors such as engagement in perceived 'high risk' sexual encounters, and past diagnosis of STIs strengthened acceptability, while others (e.g., antimicrobial resistance concerns and stigma) act as barriers. Only one study included the perspectives of healthcare workers, indicating a moderate willingness to prescribe, which would increase under governing-body endorsement.

**Data availability statement:** All data are included in the manuscript and its Supporting Information.

**Funding:** This research has been supported by a grant from the Queensland Sexual Health Research Fund financed by the Queensland Department of Health and administered by the Australasian Society for HIV, Viral Hepatitis and Sexual Health Medicine. The funder had no role in the study design, data collection and analysis, decision to publish, or preparation of the manuscript.

**Competing interests:** The authors have declared that no competing interests exist.

## Discussion

Overall, while there is some promise of STI PrEP acceptability among GBMSM, vast gaps in knowledge remain. Knowledge transfer and feasibility and, hence, the sustainability and capacity needed for the success of STI PrEP is yet to be examined and understood. However, for STI PrEP to be successfully adopted, it is essential not only to assess its acceptability and feasibility but also to focus on knowledge transfer. Knowledge transfer is a dynamic and iterative process, involving the synthesis, dissemination, exchange, and application of knowledge in an ethically sound manner. This process supports the improvement of health outcomes, strengthens healthcare systems, and ensures that healthcare interventions, such as STI PrEP, are effectively understood and implemented by both healthcare providers and at-risk populations. Similarly, the perspectives of populations beyond GBMSM have been omitted, and there is little understanding of the impact of their differing socio-cultural contexts around sex-related behaviour and Western pharmaceutical healthcare interventions on their acceptance and uptake.

## Conclusion

Further research into acceptability, feasibility and knowledge transfer among diverse high-risk groups, healthcare professionals, and policymakers is necessary to create a strong foundation for implementing STI PrEP.

## Introduction

Bacterial sexually transmittable infections (STIs) are having a resurgence in many countries, such as Australia, placing a major burden on multiple populations at heightened risk (e.g., gay and bisexual men and other men who have sex with men (GBMSM), Indigenous Australians, young people) and the health system more broadly [1,2]. This resurgence suggests that current prevention approaches and treatment methods may not sufficiently address this growing issue and that broader population-based innovative models of care may be required. Recently, HIV pre-exposure prophylaxis (PrEP) practices have provided a model potentially transferable to managing bacterial STIs [3]: STI PrEP. However, implementing a new model requires holistic knowledge and understanding of such an intervention's feasibility and potential acceptance, uptake, and adoption in patient populations and among clinicians [4].

The significant rise in incidence and subsequent morbidity from bacterial STIs is a result of numerous factors, such as a reduction in the use of condoms partly attributable to the efficacy and use of PrEP as biomedical HIV prevention [4,5] and a decreased fear of pregnancy from increased accessibility to contraception [6]. Furthermore, these risks are potentially increased by changes to sexual risk-taking behaviour promoted by contemporary dating and 'hook-up' culture prevalent amongst young people [1,7]. Many bacterial STIs may be asymptomatic, creating challenges in identification and transmission control among individuals not seeking regular testing and treatment, leading to longer-term negative health outcomes (e.g. infertility). Bacterial STIs can lead to an array of severe long-term health issues such as pelvic inflammatory disease, infertility, tubal pregnancies [8]; chronic epididymitis inflammation [9]; increased cancer risk; disseminated gonococcal infection; damage to organs, blood vessels and joints [10], and increased risk of community transmission and co-existing infections of HIV and hepatitis [11,12] and potentially hospitalisation and associated additional intensive treatments (e.g. intravenous antibiotics). Additionally, a lack of

prompt diagnosis and treatment can have significant implications for onward transmission of infection and associated sequelae [13]

Financial and time burdens to the health system attributed to bacterial STIs are exacerbated by current and insufficient models of care [14] and which, in turn, constrain health service optimisation. Current health models in many countries require that patients book appointments for screening, await laboratory test results, and follow a treatment regimen each time they test positive for an STI. This strategy is time-consuming and costly, and may be unsustainable in the face of an increasing incidence of bacterial STIs — they also present a barrier to consumers accessing medical care, resulting in an increased risk of re-infection [15] —particularly among 'harder to reach' and more vulnerable subgroups who are typically at higher risk of STIs (e.g., young adults, those who have experienced trauma, MSM and others living with intersectionality or socioeconomic disadvantage/greater social determinants of health) [1,16].

Traditional strategies to prevent bacterial STIs include promoting condom usage, and frequent screening and subsequent treatment in high-risk groups. However, prevailing attitudes of indifference or inevitability of contracting bacterial STIs in high-risk groups [17,18], have lessened the effectiveness of these strategies [3,4,19]. Partner notification methods (i.e., 'contact tracing') and accelerated/expedited partner treatment models are examples of individual-level preventative practice becoming more common [20,21]. Antibiotics are prescribed where the pre-test probability of infection is high such as due to a known partner diagnosis, prior to laboratory-based diagnosis or in the presence of known signs and symptoms [22]. While this reduces the need for all potentially exposed partners to be tested and diagnosed prior to treatment, this model of care requires at least one partner to consult a clinician, await test results, and return for a follow-up prescription. For populations where STI incidence and prevalence are high, this method of presumptive treatment continues to demand an ongoing cycle of appointments, testing and treatment. To reduce the inequitable and disproportionate burden on the high-risk groups, population-based STI PrEP of bacterial STIs has been proposed such as through a consensus statement on this approach in gay and bisexual men in Australia [23]. Population-based prevention of bacterial STIs has the potential to significantly reduce the time and financial costs for individuals and the health system present in current STI care model [4]. However, the main concerns associated with this model of care is a risk of increased antibiotic resistance resulting from long-term antibiotic use and side effects associated with antibiotics as well as stigmatisation of STI PrEP and inconvenience of taking medication a regular basis including associated adherence [17,24]. However, dermatological treatments for acne and other health issues have successfully implemented long-term low-dose antibiotics and are largely utilised treatment methods [25]. As such, STI PrEP may have potential for the mitigation of STI spread.

A recent randomised trial conducted in the USA with MSM reported the efficacy of a daily dosing regimen of doxycycline PrEP, with a US randomised controlled trial among MSM living with HIV recording a 73% reduction in syphilis, chlamydia, and gonorrhoea incidences among the treatment group [26]. Other modes of PrEP (such as period PrEP) have also proven to be effective [27,28]. While the efficacy of these models of care is demonstrated in clinical trials, translation to population-based implementation requires further knowledge [4]. Essential to the wider adoption of STI PrEP is an understanding of the attitudes and beliefs guiding treatment use in and prescription of antibiotic STI PrEP. Existing insights [3] show early indications of potential patient uptake upon trusted recommendation, interest in trialling, and willingness to use doxycycline PrEP in a patient population (specifically MSM). Further developing these understandings will provide valuable insight into the potential benefits of a population-based antibiotic STI PrEP, building the knowledge necessary for implementation of such an intervention.

Currently, there is limited available research on the acceptability and feasibility of STI PrEP. Thus, the aim of this review is to systematically investigate the *extent to which existing research examines the acceptability and feasibility of STI PrEP and models of care for bacterial STIs among high-risk populations and clinicians. STI PrEP, for the purpose of this systematic review, is defined as the use of antimicrobial therapy in asymptomatic individuals who are at risk of the primary acquisition of bacterial sexually transmitted infections, administered prior to a potential exposure.*

## Materials and methods

This review adopted published guidelines for narrative reviews. A PRIMSA checklist is provided as supplementary material (see S1 File) [29]. A protocol for the review has been registered with the international prospective register of systematic reviews by the National Institute for Health and Care Research (Protocol number: CRD42023455250). All materials used for the review can be found in this report and the supplementary materials.

### Search strategy

A comprehensive search strategy was developed involving terms relating to the acceptability and feasibility of STI PrEP approaches; the development of the search strategy involved a librarian specialised on systematic literature searches as well as PhD-level epidemiologists, pharmacists and social scientists. Searches were conducted in October 2024 across six databases: PubMed, Medline, EMBASE, CINAHL, PsycInfo, Health Systems evidence and Health evidence.org. These databases were selected based on their extensive coverage of health and health-related research. Three concepts were used in the search:

- Concept 1: Populations (e.g., patient).

- Concept 2: Disease and intervention (e.g., STI, antibiotic).

- Concept 3: Outcome (e.g., acceptability of health care).

A full example search strategy can be found in the supplementary material (see S2 File).

Only articles published from 2012 onwards were included in the searches consistent with the publication of the *interim guidelines concerning HIV pre-exposure prophylaxis* by the Centres for Disease Control and Prevention [30]. This cut-off date has been chosen to align with this important development in the field of pre-exposure prophylaxis for HIV that in the aftermath demonstrated a significant shift in approaches to treatment, sexual risk behaviours and the acceptance and update of pre-exposure prophylaxis, particularly in high-risk groups such as men who have sex with men. Furthermore, this cut-off ensures that included studies are consistent with contemporary understandings of PrEP. Results of the database searches were exported to Covidence systematic review software with title and abstract reviews conducted independently by two members of the research team. Discrepancies were resolved by a third author. Full-text screening was then conducted by two members of the research team, with conflicts surrounding study relevance adjudicated and resolved through consultation with a third author. Upon completion of full-text screening, reference lists of all articles eligible for extraction and other noted relevant review articles underwent title and abstract screening, and then full-text screening and approval to ensure all eligible articles that may not have been returned by the database search strategy were included to enable comprehensive analysis. Data extraction and risk-of-bias assessments were completed by two members of the research team, with cross checking and deliberation of discrepancies done in collaboration with the research team.

### Inclusion and exclusion criteria

Articles were deemed relevant if they were

1. Peer-reviewed.

2. Examined actual or theoretical usage of antibiotics as STI PrEP for at least one of the most commonly diagnosed bacterial STIs (*chlamydia, Gonorrhoea, syphilis, mycoplasma genitalium, donovanosis, chancroid*).

3. Reported on any aspect of acceptability and/or feasibility of the STIP PrEP treatment approach and/or knowledge transfer, and

4. Of any study design that involved either primary or secondary data.

Studies were excluded if they examined paediatric populations (below 14 years), only reported on treatment approaches for fully diagnosed bacterial STIs (e.g., partner treatment), only reported on the effectiveness of the antibiotic treatment, were written in a language other than English or did not involve any of the bacterial STIs specified in the inclusion criteria. A list of all articles excluded in the full-text review with reasons can be found as supplementary information (see S3 Table).

This review specifically examined STI PrEP models that employed the prescription of antibiotics to prevent bacterial STIs completely independent of specific exposure risk events (e.g., post-exposure prophylaxis; PEP) and any level of symptom assessment or ab-confirmed diagnosis (e.g., partner notification, expedited/accelerated treatment).

For this review, the information regarding the acceptability of STI PrEP was guided by Sekhon et al.'s [31] definition of *"a multi-faceted construct that reflects the extent to which people delivering or receiving a healthcare intervention consider it to be appropriate, based on anticipated or experienced cognitive and emotional responses to the intervention."* Investigation of feasibility will work under the definition of *"the practicality and adequacy of the logistics required for delivering interventions."* [32]. Additionally, articles were determined relevant based on knowledge transfer and implementation under the definition of Straus et al. [33]; *"a dynamic and iterative process that includes the synthesis, dissemination, exchange and ethically sound application of knowledge to improve health, provide more effective health services and products, and strengthen the health care system."* Results are provided narratively without a meta-analysis due to heterogeneity in the study designs and diversity of the included populations. Each included study was assessed for risks of bias in the study design to assess the certainty of the study's findings using the mixed-methods appraisal tool (MMAT) [34]. The MMAT tool has been chosen given the inclusion of different study designs in this systematic review. Detailed results of the MMAT tool can be found in the supplementary information (see S4 Table).

The descriptive details and key findings of the included papers were entered into an article matrix using Excel. This method, as described by Popenoe & Langius-Eklöf [35], was used to facilitate data extraction. A narrative synthesis approach was then utilised to group similar key findings in line with the research aims [36]. No missing data were identified by the authors or reported in the included studies.

## Results

A total of 10,80 citations were retrieved and imported to Covidence, after 122 duplicates were removed, 958 citations were included in title and abstract screening. Title and abstract screening resulted in exclusion of 910 citations leading to 48 citations being screened full-text. Following completion of full-text and reference list screening, eight studies from eight publications were eligible for extraction and included in the review as can be seen in Fig 1 [37–44].

A breakdown of the characteristics of included studies is provided in the supplementary materials (see S5 Table). *No studies were identified investigating knowledge transfer or feasibility.* All included studies investigated aspects of acceptability in a total of 6,542 participants. Of the eight studies included, seven were conducted in high-income countries, three were conducted in Australia [37,39,42], two were conducted in each the United States [41,44] and Canada [38,40], and one in China [43]. All eight studies focused on sexual minority men using different terminology (e.g., gay and bisexual men or men who have sex with men) with one study also involving healthcare providers from the U.S. with prescribing authority [44]. Four studies used cross-sectional surveys [37,38,43,44], two studies used qualitative interview [39,40], and one each using an observational cohort study [41] and one applying a mixed-methods approach [42,45]. The quality of all studies has been analysed using the MMAT tool; research questions were clearly formulated in all studies and the data collected was appropriate to address the research questions. Detailed information on all domains can be found in S1 and Table 1.

In Arapali et al's [37] cross-sectional study among a sample of 1,347 HIV pre-exposure prophylaxis experienced (from the EPIC_NSW PrEP implementation project) Australian gay and bisexual men enrolled in New South Wales, Australia, more than half the participants (54.3%, n = 732) indicated that they were willing to use STI-PrEP. These findings are consistent with Park et al's [44] cross-sectional survey from the United States, which showed a slightly higher acceptance with 67.5% (n = 143) among men who have sex with men as well as 52.7% (n = 1,104) in another study among gay men from Australia [45]. Zhang et al's cross-sectional study from China [46] presented participants with a choice between PEP (post-exposure prophylaxis) and PrEP (pre-exposure prophylaxis) mode of doxycycline delivery for syphilis infections in which the majority of participants preferred PEP over PrEP mode (67.8%, n = 415). This finding is consistent with a Fusca et al's [47] cross-sectional survey of gay and bisexual men in Canada, in which participants also showed a stronger preference towards PEP delivery of doxycycline rather than as PrEP with 60.1% (n = 268) of participants indicating willingness to use PEP compared to 44.1% (n = 197) willing to use PrEP. All qualitative interview studies included in this review came to the conclusion that there is interest in STI PrEP with one study showing it to be among the most popular interventions among men who have sex with men [41], while participants in another study expressed cautious optimism for this type of intervention [17] or showed a generally high level of interest [40].

Park et al's [44] cross-sectional survey from the United States also included healthcare workers with varying levels of acceptability to prescribe depending on the context with 43.3% (n = 44) being generally willing to prescribe this type of medication. However, willingness would increase to 89.5% (n = 68) if this type of treatment would be endorsement by the Centres for Disease Prevention and Control.

Five studies reported on a number of factors that impact on acceptability or willingness to use this type of intervention. Most commonly identified factors that positively impacted acceptability/willingness were larger numbers of sexual partners or engagement in perceived 'high-risk' sexual encounters or generally higher perceived personal risk [37,41,47], or being engaged in chemsex, including the use methamphetamine (also known as crystal meth, an illicit psychoactive substance that is commonly involved in chemsex) [17,37], consciousness about avoiding STIs [37], past diagnoses of STIs [17,37,44,47] or being on HIV PrEP [37,44,47]. Other potential factors were also analysed in some of the studies; for example, Park et al [44] found no statistically significant differences were found between sexual orientations or living area in their cross-sectional study from the United States; however, a significant difference was identified between for race with African-American (74.1%, n = 20) and white (73.8%, n = 62) participants showing a generally higher acceptance than Asian participants

Acceptability and feasibility of population-based presumptive treatment approaches to bacterial sexually transmissible infections

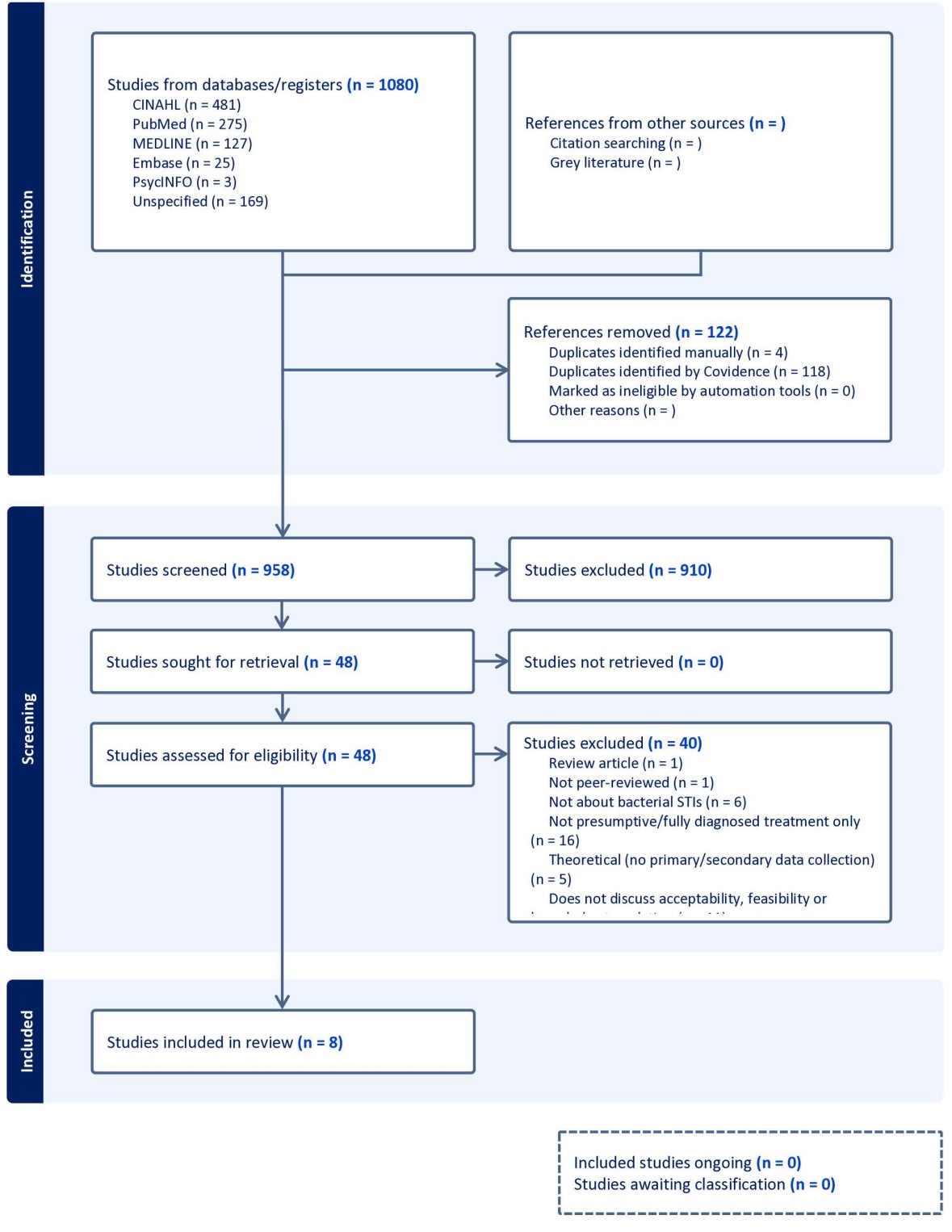

**Fig 1. PRISMA flowchart.**

**Table 1. Areas for future research.**

- Global perspectives. Conduct further studies in non-Western/non-Eurocentric countries.
- Healthcare provider perspectives. Further studies looking at the perspectives of healthcare providers are requires to understand their perspectives on this topic including willingness to prescribe.
- Diverse populations. Research in this area must expand into further populations, especially those at higher risk of bacterial STIs such as young women, sex workers, culturally and linguistically diverse people as well as Indigenous peoples.
- Stigma aWnd social perceptions. Future research should aim at exploring the role of stigma and social perceptions in the acceptability of presumptive treatment approaches.
- Knowledge transfer. A significant gap in the current research requires future research to aim at understanding knowledge transfer concerning this treatment in priority populations.

(50.0%, n = 15). The study similarly identified that a recent history of sexually transmissible infections/diagnoses as well as concerns about contracting STIs and currently being on HIV PrEP led to a higher acceptance.

A range of barriers were identified, particularly concerns around antimicrobial resistance and side effects as a result of frequent and broad use of antibiotics [17,40,44]; this concern was shared among healthcare professionals [44]. Other barriers included costs [40], lack of education around this type of treatment or generally limited sexual health literacy [47] and stigma associated (e.g., association with promiscuity) with the uptake of this type of treatment [17,30].

## Discussion and conclusion

A systematic review of the existing body of literature was conducted to understand the extent of the current global research investigating and reporting findings on the feasibility and acceptability of presumptive treatment approaches to prevent bacterial STIs. Only eight studies met the criteria of examining actual or theoretical use of antibiotics as a treatment for common STIs, and reported on this via primary or secondary data [37–44]. Further, while all included identified studies examined acceptability, none examined knowledge transfer or feasibility, indicating a need for more expansive approaches to investigating the long-term sustainability and capacity for such an intervention to succeed. The studies included in this review primarily focussed on GBMSM in high-income countries who were already familiar with HIV PEP and PrEP approaches. The recency and homogeneity of the study population target groups could be explained by the focus of most research within the past decade on PEP and PrEP for HIV in these groups [48–51].

Due to the near-exclusive examination of GBMSM in high-income countries to assess acceptability of PEP doxycycline, a clear gap in knowledge has been identified regarding the responsiveness of such a program to women, sex-workers, as well as members of CALD communities and Indigenous groups. Although one included study was conducted in China, the core findings were a preference for PEP compared with PrEP approaches to STI transmissions. While these findings were mirrored in the Canadian study also focussed on GBMSM [38], further studies with a high acceptability of PrEP approaches, did not discuss PEP at all, making it difficult to understand preferences if participants are not given a choice between options. Overall, further research is required in non-Eurocentric countries. Similarly, STI PrEP may have the potential for significant benefit in other high-risk contexts where there is also well-established higher incidence of STIs and greater potential scope of impact for new STI prevention and treatment models (e.g., STI PrEP). For example, in prisons [52] and correctional environments [53]; in promoting harm reduction amongst overseas travellers [54,55], migrants, CALD communities [56–58] and university students (including international students and overseas born domestic students; [59–61].

Sexual minority men (GBMSM) were found for the most part to have a high approval rate of an preventative approach to STIs via a low-dose antibiotic ongoing treatment, especially if they were already on HIV PrEP, had regular sexual relationships with different partners whom they perceived to be 'high risk', had engaged in chemsex, or who had previous diagnoses of STIs. These findings have important implications for how research shapes future questions regarding acceptability in different demographic groups wherein in the social and cultural meanings around sex-related behaviour differ widely. For example, while sexual minority men who engaged in higher-risk behaviour relating to STI transmission expressed an overall positive response to the idea of a doxycycline PEP, this same finding may not be transferable to a group such as heterosexual women, for whom any behaviour involving a high number of partners or frequent sexual intercourse may be associated with negative social judgement and stigma [62]. This might also be due to the fact that there are generally more open cultures in talking about sexual health in sexual minority groups [63]. Women have also been historically shown to be more cautious than men when it comes to a range of behaviours, including the acceptance of medical interventions such as new vaccines, and lower participation rates in pharmaceutical clinical trials [64]. While women may engage in sex with a high number of partners, they are unlikely to be willing to disclose these numbers or admit these risks due to stigma, and overall, they are unlikely to record the same number of partners as sexual minority men [65]. This, combined with their increased cautiousness around medicinal and pharmaceutical intervention may make them a hard-to-reach group for an intervention such as this.

Interestingly, while heterosexual women, as well as sexual minority women, may be more risk averse in both sexual behaviours and acceptance of medicinal and pharmaceutical trials than sexual minority men, these groups do have one thing in common: they have both contributed to the increase in STI spread globally due to rapid improvements in contraceptive devices for women, and the large success rates of PEP and PrEP for HIV respectively, lowering the perceived need of condoms. In countries such as Australia in particular, STIs such has syphilis have seen dramatic increases [2]. Researchers are speculating from these recent findings that this is due to the fact that women are now far less likely to fear an unwanted pregnancy, and sexual minority men no longer fear HIV and AIDS [66]. It has been suggested that risk compensation related to the increasingly widespread use of both HIV PrEP and birth control may contribute to rising STI rates, particularly through reduced condom use. While the concept of risk compensation is controversial, especially given the longstanding availability of birth control without historic STI surges, recent studies have shown that individuals on HIV PrEP are more likely to engage in condomless sex, thus increasing their exposure to bacterial STIs [3,5]. This is consistent with risk compensation theory, which argues that a perceived reduction in one risk (e.g., pregnancy or HIV) may lead to increased exposure to other risks. Furthermore, rising STI rates are multifactorial, and other factors, such as increased testing, evolving sexual networks, and antimicrobial resistance, must also be considered [15,24]. While risk compensation provides a plausible explanation, it is important to acknowledge that it operates alongside various other complex sociocultural and healthcare dynamics contributing to the resurgence of bacterial STIs.

Lower socioeconomic groups – both within and between countries globally – as well as African American and Indigenous Australians, have also expressed scepticism and fear of many of the suggestions made by a wealthy, white-dominated healthcare system that holds inherent power structures and, for the most part, embodies institutionalised racism [67,68]. Lower socioeconomic groups, especially in high-income countries, are usually the last to adopt messages within health promotion and education campaigns, including messages regarding SIDS prevention smoking cessation, increased fruit and vegetable consumption,

screening test, and more recently the COVID-19 vaccine [69] for a variety of reasons such as limited access to resources including a lack of health insurance, lower health literacy as well as competing life stressors and the influence of social norms that may not prioritise health-promoting behaviours as priority behaviours [70]. Additionally, socioeconomic disparities often exacerbate challenges in understanding, accessing, and acting on health-related information, further delaying the adoption of positive health behaviours. This may well extend and apply to a preventive treatment for STIs if recommended by a GP or via a traditional health promotion campaign. Black American and Indigenous groups demonstrate similar mistrust of health communication and promotion coming from healthcare systems that have historically not adhered to practices of cultural safety, nor have they resulted in a reduction of health inequalities for these groups globally [71]. Specifically, many groups in Africa refused, and continue to refuse condom use due to suspicions that this is an attempt by white authorities to wipe out their race or render them powerless and other beliefs that impact STI/HIV prevention [72] Social and structural barriers pertaining to race and class that have previously applied to the acceptance and uptake of several health campaigns led by medical and pharmaceutical authorities historically may indeed play out in the case of a proposed treatment such as this one; especially one that has implications for such an intimate aspect of their lives.

Overall, this study concluded that early findings into the acceptability of doxycycline PrEP is likely to be high in sexual minority men in high income countries. The acceptability, knowledge and feasibility need further research both in these groups and in other identified at-risk groups. Potential hesitancy in some at-risk groups combined with a rapid rise in common STIs, points to the fact that such approaches need to be tested with high priority. Both hesitancies to adhere to risk reducing behaviours in STIs combined with their consequent increase holds implications for other groups such as sex workers and any population likely to affect these trends in the near future. Various initiative could enhance STI PrEP uptake and reach to various populations, such as bilingual community health workers in CALD communities [73]; mobile outreach to rural and remote communities [74], and addressing stigma among health professionals to make accessing health services less stigmatising re STI/HIV and among priority sub-groups [75,76]. However, further research int these areas is required; an overview of potential areas for the future research agenda can be found in Table 1.

## Supporting information

**S1 File. PRIMSA statement checklist.**
(DOCX)

**S2 File. Full search strategy.**
(DOCX)

**S3 Table. Studies excluded in full-text review.**
(DOCX)

**S4 Table. MMAT checklist.**
(XLSX)

**S5 Table. Characteristics of included studies.**
(DOCX)

## Acknowledgments

We would like to acknowledge Diane Heart and Claire Moran for their contributions to this work.

## Author contributions

**Conceptualization:** Julie-Anne Carroll, Amy B. Mullens, Philip RA Baker, Daniel Demant.

**Data curation:** Daniel Demant.

**Formal analysis:** Sarah Warzywoda, Meika Stafford, Daniel Demant.

**Funding acquisition:** Julie-Anne Carroll, Amy B. Mullens, Philip RA Baker, Daniel Demant.

**Investigation:** Julie-Anne Carroll, Daniel Demant.

**Methodology:** Julie-Anne Carroll, Amy B. Mullens, Philip RA Baker, Faye McMillan, Jacintha Manton, Daniel Demant.

**Project administration:** Julie-Anne Carroll.

**Writing – original draft:** Julie-Anne Carroll, Sarah Warzywoda, Meika Stafford, Daniel Demant.

**Writing – review & editing:** Julie-Anne Carroll, Amy B. Mullens, Sarah Warzywoda, Philip RA Baker, Faye McMillan, Jacintha Manton, Daniel Demant.

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
