## [Decision Letter · Decision Letter 0]

4 Sep 2024

PONE-D-24-09738Acceptability and feasibility of presumptive treatment for bacterial STIs: A systematic reviewPLOS ONE

Dear Dr. Demant,

Thank you for submitting your manuscript to PLOS ONE. After careful consideration, we feel that it has merit but does not fully meet PLOS ONE’s publication criteria as it currently stands. Therefore, we invite you to submit a revised version of the manuscript that addresses the points raised during the review process.

We look forward to receiving your revised manuscript.

Kind regards,

Carlos Miguel Rios-González, Ph.D

Academic Editor

PLOS ONE

Journal Requirements: 

2. Thank you for submitting the above manuscript to PLOS ONE. During our internal evaluation of the manuscript, we found significant text overlap between your submission and previous work in the [introduction, conclusion, etc.].

Please revise the manuscript to rephrase the duplicated text, cite your sources, and provide details as to how the current manuscript advances on previous work. Please note that further consideration is dependent on the submission of a manuscript that addresses these concerns about the overlap in text with published work.

[If the overlap is with the authors’ own works: Moreover, upon submission, authors must confirm that the manuscript, or any related manuscript, is not currently under consideration or accepted elsewhere. If related work has been submitted to PLOS ONE or elsewhere, authors must include a copy with the submitted article. Reviewers will be asked to comment on the overlap between related submissions (http://journals.plos.org/plosone/s/submission-guidelines#loc-related-manuscripts).]

We will carefully review your manuscript upon resubmission and further consideration of the manuscript is dependent on the text overlap being addressed in full. Please ensure that your revision is thorough as failure to address the concerns to our satisfaction may result in your submission not being considered further.

3. As required by our policy on Data Availability, please ensure your manuscript or supplementary information includes the following: 

Reviewers' comments:

Reviewer's Responses to Questions

**Comments to the Author**

1. Is the manuscript technically sound, and do the data support the conclusions?

Reviewer #1: Yes

2. Has the statistical analysis been performed appropriately and rigorously? 

Reviewer #1: N/A

3. Have the authors made all data underlying the findings in their manuscript fully available?

Reviewer #1: Yes

4. Is the manuscript presented in an intelligible fashion and written in standard English?

Reviewer #1: No

5. Review Comments to the Author

Reviewer #1: Summary of review

The authors are reviewing alternative methods of STI treatment and prevention which is needed in the era of syndromic management and limited testing. This is an important topic that needs highlighting. Below are suggestions for increasing the clarity of this work.

Overall

1. The authors can remove infection from the term “STI infection” since the I in STI refers to infection.

2. Please review this manuscript for grammar and sentence structure. Throughout there are errors causing incomprehension of sentences.

3. Please define the purpose of the study clearly (give examples of the types of presumptive treatment of interest) and then use consistent language/terms throughout the manuscript.

4. When discussing the articles, please be clear. Clearly indicate the study and at minimum include the study design and population each time a study is mentioned as readers might not be able to find mention of the study earlier in the text.

Abstract

1. Please provide more details in the Methods section. I, potentially like other readers, am surprised to see that only eight studies met the inclusion criteria for this review. Since most people only read the Abstract, this will be very useful to have more details here.

Introduction

1. The introduction is very long and contains a lot of unnecessary detail. Also, the authors do not even introduce the focus of the article, STI PrEP until page 5. All details prior to this paragraph could be significantly shortened. For example, paragraph 70-79 could be shortened to one sentence.

2. Potentially a typo on lines 95-96 – similar language used twice.

3. Likely typo 136-139 – couldn’t understand sentence.

4. I did not realize until the end of the Methods section that the focus of this review did not include STI PEP. I think one reason for this is the authors introducing so many constructs in the Introduction – I’m not sure what is Background and what is framing the purpose of the article. Another reason is potentially interchanging presumptive treatment with STI PrEP. It would be helpful for the authors to define the scope of their manuscript and review and choose one term to define this. For example, presumptive treatment needs further definition. It can mean mass treatment of a community, symptomatic treatment without a diagnosis, daily STI PrEP, etc.

Methods

1. It would be helpful for the authors to list the search terms for each concept as this will ease reproducibility.

2. Can the authors explain their rationale for beginning the STI presumptive treatment search to correlate with HIV PrEP guidelines?

3. Line 189 - Bacterial vaginosis is not an STI. It is dysbiosis and shift of normal vaginal flora. This is not a disease state that would require presumptive treatment.

4. Can the authors describe what they mean by “knowledge transfer” when this is first introduced – prior to the definition in the Methods section?

Results

1. Line 238 – cite the mixed methods study.

2. The sentence from 245-247 is confusing as written. It was initially unclear to me that the second percentage referred to a different study. It would help to also contextualize the Park study (setting).

3. The sentence from 250-253, please state the comparator group of 44.1% for clarity.

4. Lines 258-261, how did this study measure the change in healthcare workers acceptability of STI PrEP if the Park study is a cross-sectional study as stated above.

Discussion

1. Lines 294-296, the authors mention a study in China, but do not complete this thought.

2. Line 315-318, are women more cautious or have they been excluded from major studies due to fear of pregnancy? If the authors are going to make this argument, they should tease this out a bit more.

3. Line 323-325 – the authors are comparing heterosexual women and sexual minority women to “sexual minority counterparts” – this doesn’t make sense since both arms of the comparison include sexual minorities.

4. 330-334, the authors posit that risk compensation due to birth control and HIV PrEP have contributed to the rising STI rates. While this is one theory, it is controversial. Birth control has been present since the 1970’s yet STIs were not rising back then. Could the authors contextualize the idea of risk compensation and also consider other arguments for rising STIs.

5. 338-342 – A major reason for decreased uptake among people of lower SES is a lack of resources and structural barriers – ex: food deserts, lack of insurance, etc.

6. PLOS authors have the option to publish the peer review history of their article (what does this mean? ). If published, this will include your full peer review and any attached files.

**Do you want your identity to be public for this peer review?** For information about this choice, including consent withdrawal, please see our Privacy Policy .

Reviewer #1: No

---

## [Author Response · Author response to Decision Letter 1]

24 Oct 2024

Responses to reviewers are included in the attached file named "Response to Reviewer Comments".

---

## [Editor Report · Decision Letter 1]

3 Jan 2025

Acceptability and feasibility of pre-exposure prophylaxis for bacterial STIs: A systematic review

PONE-D-24-09738R1

Dear Dr. Demant,

We’re pleased to inform you that your manuscript has been judged scientifically suitable for publication and will be formally accepted for publication once it meets all outstanding technical requirements.

Kind regards,

Matthew J. Mimiaga, ScD, MPH

Academic Editor

PLOS ONE

Additional Editor Comments (optional):

We have facilitated the peer-review of your manuscript, “Acceptability and feasibility of pre-exposure prophylaxis for bacterial STIs: A systematic review.” Overall, the reviewers were enthusiastic about this review on the acceptability and feasibility of STI PrEP. All three reviewers found your paper to be of great public health interest and were enthusiastic about its publication in PLOS One. Based on their expert reviews, I am in agreement. However, the authors state in their paper that “No studies were identified investigating knowledge transfer or feasibility.” Hence the authors should remove the word “feasibility” in the title of their paper. They must also add the word “perceived” in the title and paper when referencing “acceptability,” since this review covers “perceived acceptability” among the samples.
---

## [Editor Report · Acceptance letter]

PONE-D-24-09738R1

PLOS ONE

Dear Dr. Demant,

I'm pleased to inform you that your manuscript has been deemed suitable for publication in PLOS ONE. Congratulations! Your manuscript is now being handed over to our production team.

Kind regards,

on behalf of

Dr. Matthew J. Mimiaga

Academic Editor

PLOS ONE